# From Genomics to Metabolomics: Molecular Insights into Osteoporosis for Enhanced Diagnostic and Therapeutic Approaches

**DOI:** 10.3390/biomedicines12102389

**Published:** 2024-10-18

**Authors:** Qingmei Li, Jihan Wang, Congzhe Zhao

**Affiliations:** 1Honghui Hospital, Xi’an Jiaotong University, Xi’an 710054, China; 2Institute of Medical Research, Northwestern Polytechnical University, Xi’an 710072, China

**Keywords:** osteoporosis, genomics, transcriptomics, proteomics, metabolomics, diagnostic biomarkers, therapeutic interventions

## Abstract

Osteoporosis (OP) is a prevalent skeletal disorder characterized by decreased bone mineral density (BMD) and increased fracture risk. The advancements in omics technologies—genomics, transcriptomics, proteomics, and metabolomics—have provided significant insights into the molecular mechanisms driving OP. These technologies offer critical perspectives on genetic predispositions, gene expression regulation, protein signatures, and metabolic alterations, enabling the identification of novel biomarkers for diagnosis and therapeutic targets. This review underscores the potential of these multi-omics approaches to bridge the gap between basic research and clinical applications, paving the way for precision medicine in OP management. By integrating these technologies, researchers can contribute to improved diagnostics, preventative strategies, and treatments for patients suffering from OP and related conditions.

## 1. Introduction

Osteoporosis (OP) is a pervasive skeletal disorder characterized by decreased bone density and strength, along with structural deterioration, leading to increased bone fragility and a heightened risk of fractures. This condition, often termed the “silent disease” due to its asymptomatic nature until a fracture occurs, represents a significant public health concern worldwide [1,2]. The prevalence of OP is particularly high among postmenopausal women, though it also affects men and younger individuals, especially those with specific risk factors such as chronic glucocorticoid use, certain endocrine disorders, or lifestyle factors like smoking and inadequate physical activity. A systematic review and meta-analysis reported that the global prevalence of OP is 18.3%, with a significantly higher prevalence in women (23.1%) compared with men (11.7%). The highest prevalence was reported in Africa, where 39.5% of the population is affected. OP incidence is projected to increase with the aging population, making it a pressing issue for healthcare systems [3]. Despite abundant sunlight, the Middle East and Africa have some of the highest rickets rates globally. This could be due to the widespread occurrence of vitamin D deficiency in these regions, which may also contribute to OP [4,5,6]. Additionally, the elevated prevalence of OP in Africa may be attributed to the continent’s aging population, increased survival rates among individuals with human immunodeficiency virus (HIV), and limited access to screening and diagnostic tools, such as bone densitometry. Additionally, lack of awareness about OP, socio-economic barriers, and inadequate epidemiological data further contribute to the challenge of accurately quantifying and addressing the disease [7].

The impact of OP extends beyond the physical consequences of fractures (also known as fragility fractures), which are often associated with pain, disability, and decreased quality of life. OP fractures, particularly in postmenopausal osteoporosis (PMOP), significantly increase disability, morbidity, and mortality, yet the treatment rates remain low [8,9]. A study in Denmark found that only 16.7% of patients with recent hip fractures received anti-OP treatment, and just 9.8% underwent a dual-energy X-ray absorptiometry (DXA) scan post-fracture, highlighting a critical gap in follow-up care [10]. This global issue underscores the need for early intervention and routine DXA referrals to improve patient outcomes. Vertebral fractures, another common manifestation, can lead to chronic pain, height loss, and deformities such as kyphosis, further contributing to the disease’s burden [11,12]. The economic costs associated with OP are substantial, encompassing not only direct medical expenses for fracture treatment and rehabilitation but also indirect costs related to long-term care and lost productivity. A systematic review found that treating OP fractures incurs direct annual costs ranging from USD 5000 to USD 6500 billion in regions like Canada, Europe, and the USA. Effective prevention and management strategies could substantially reduce these financial burdens on healthcare systems [13].

In recent years, the field of OP research has been revolutionized by the advent of omics technologies, which provide powerful tools for exploring the complex molecular underpinnings of this disease. These technologies have been particularly impactful because they allow researchers to delve deeper into the molecular mechanisms that drive OP. By enabling the analysis of gene expression, protein interactions, and metabolic changes, omics approaches uncover intricate relationships between biological pathways, offering insights that were previously inaccessible. The term “omics” refers to the collective technologies used to explore the roles, relationships, and actions of the various types of molecules that make up the cells of an organism [14]. The various omics technologies, including but not limited to genomics, transcriptomics, proteomics, and metabolomics, each provide distinct layers of biological information [15,16]. Genomics, for example, involves the study of an organism’s complete set of DNA, including all of its genes, and has been pivotal in identifying genetic variants associated with bone mineral density (BMD) and fracture risk [17,18]. Transcriptomics, which focuses on the analysis of RNA transcripts, has shed light on gene expression patterns and regulatory mechanisms that may influence bone metabolism [19,20]. Proteomics, the large-scale study of proteins, has enabled the identification of protein signatures and signaling pathways involved in bone remodeling, while metabolomics, the study of small molecules or metabolites, has provided insights into the metabolic changes associated with bone health [21,22,23,24]. Together, these omics approaches have vastly expanded our understanding of OP, revealing that it is not merely a disease of aging or calcium deficiency but a complex condition influenced by a multitude of genetic, epigenetic, and environmental factors.

The integration of omics data—often referred to as multi-omics—holds the promise of a more comprehensive understanding of health and disease, enabling the development of personalized medicine approaches tailored to an individual’s unique molecular profile [25,26,27,28]. By harnessing the power of multi-omics, researchers and clinicians are better equipped to identify novel biomarkers for early diagnosis, predict disease progression, and develop targeted therapeutic strategies that address the underlying molecular causes of OP [29,30,31]. In this review, we will explore the evolving role of omics in understanding OP, from the identification of genetic risk factors and the elucidation of gene expression profiles to the discovery of protein biomarkers and the characterization of metabolic alterations. We will also discuss the integration of multi-omics data and its potential to transform the diagnosis and treatment of OP, paving the way for a new era of precision medicine in bone health.

## 2. Genomics in Osteoporosis: Unraveling the Genetic Basis

### 2.1. Genetic Predispositions and Risk Factors

OP is a multifactorial disease with a strong genetic background. Studies have suggested that genetic factors play a crucial role in determining BMD, the key determinant of OP risk [32]. BMD is critical because it is a key determinant of bone strength, with lower BMD associated with an increased risk of fractures. Because BMD reflects both bone density and quality, it serves as an essential predictor of fracture risk, making its understanding vital for diagnosing and managing OP. Family and twin studies estimate that up to 60–80% of the variance in BMD is attributable to genetic factors, highlighting the significance of heredity in OP [33]. Over the past few decades, researchers have identified numerous genes that contribute to bone health, many of which are involved in bone formation, resorption, and the regulation of calcium and phosphate metabolism. Key genes implicated in OP include those encoding for collagen type I alpha 1 (COL1A1), which is a major component of the bone matrix, and the vitamin D receptor (VDR), which is critical for calcium absorption and bone mineralization [34,35,36,37]. Variants in these genes have been associated with BMD changes and fracture risk. Additionally, genes involved in the Wnt signaling pathway, such as LRP5 and SOST, have been identified as important regulators of bone formation [38,39,40]. Mutations or polymorphisms in these genes can lead to altered bone mass and structure, predisposing individuals to OP.

Beyond single gene effects, the interplay of multiple genetic factors, along with environmental influences, shapes an individual’s overall risk of developing OP. Polygenic risk scores, which aggregate the effects of numerous genetic variants across the genome, have been developed to predict an individual’s susceptibility to OP. These scores, while still in the early stages of clinical application, represent a promising tool for identifying individuals at high risk for the disease, allowing for earlier interventions and personalized prevention strategies.

### 2.2. Genome-Wide Association Studies (GWASs)

The advent of genome-wide association studies (GWASs) has revolutionized our understanding of the genetic basis of OP. GWASs allow researchers to scan the entire genome for common genetic variants, known as single nucleotide polymorphisms (SNPs), that are associated with complex traits such as BMD and fracture risk [41,42,43]. By pinpointing genetic variations that influence bone metabolism, SNPs help researchers map out the biological processes governing osteoblast and osteoclast function, which is invaluable for understanding disease mechanisms and developing targeted therapies for OP. This approach has led to the identification of hundreds of loci linked to OP, many of which were previously unknown. The GWAS analysis has uncovered more than 1000 genetic associations, marking a significant leap in our understanding of OP and related conditions [44]. In 2021, Xiaowei et al. reviewed the significant strides made in OP research through GWASs over the past 12 years [45]. It details the discovery of numerous genetic loci associated with BMD, OP, and osteoporotic fractures, highlighting key genes such as LRP5, RANKL, ESR1, BMP4, and WNT16 that play crucial roles in bone metabolism and endochondral ossification. They also explore the use of Mendelian randomization to establish causal links between clinical risk factors and OP outcomes [45].

Major findings from GWASs have provided new insights into the biological pathways involved in bone metabolism. For example, several loci identified through GWAS are related to the regulation of bone cell activity, including osteoblasts (cells responsible for bone formation) and osteoclasts (cells responsible for bone resorption). A study led by Madison et al. integrates BMD GWAS results with single-cell RNA-sequencing to identify and prioritize candidate genes influencing age-related bone loss. By focusing on variant nearest-neighbor (VNN) genes and their upregulation in aged mice, the research highlights the role of specific cell types in bone and marrow. The findings suggest that many VNN genes, particularly those involved in extracellular matrix (ECM) deposition by osteoblasts, could be key in maintaining BMD and offer novel avenues for OP intervention [46]. A GWAS focusing on lip BMD in Europeans has identified significant genetic signals associated with OP susceptibility, with a particular emphasis on immune pathways [47]. The study, which utilized gene-based and pathway-based analyses, discovered that three immune-related KEGG pathways—the T cell receptor signaling pathway, complement and coagulation cascades, and the intestinal immune network for IgA production—were significantly associated with BMD, suggesting a substantial role of immune mechanisms in OP risk [47]. Moreover, GWASs have uncovered genetic variants that interact with environmental factors, such as diet and physical activity, to influence OP risk. A GWAS within the Boston Puerto Rican Osteoporosis (BPROS) study has identified four genetic variants associated with OP and two with BMD in a Hispanic cohort of older adults. The research also discovered significant gene–diet interactions, particularly between SNPs and sugar-sweetened beverages (SSBs), suggesting that dietary factors can modulate the genetic risk of OP and emphasizing the potential for personalized nutrition strategies in the prevention and early detection of the disease [48]. Gene–environment interactions (G × E) [49] underscore the complexity of OP as a disease, where genetic predispositions are significantly influenced by lifestyle factors such as diet, exercise, and smoking. For example, regular physical activity can mitigate genetic risk by enhancing bone formation, while inadequate intake of calcium and vitamin D may exacerbate genetic vulnerabilities, leading to lower BMD and increased fracture risk [50,51]. These interactions highlight the need for comprehensive, personalized approaches to OP prevention and management that consider both genetic background and modifiable lifestyle factors.

### 2.3. Epigenetic Modifications in Osteoporosis

In addition to genetic variations, epigenetic modifications play a crucial role in the regulation of gene expression and are increasingly recognized as important contributors to OP. Epigenetics refers to heritable changes in gene expression that do not involve alterations in the DNA sequence itself but are instead mediated by mechanisms such as DNA methylation, histone modifications, and non-coding RNAs [52,53,54,55], as summarized in Figure 1.

#### 2.3.1. DNA Methylation

DNA methylation, the addition of a methyl group to cytosine residues in DNA, can lead to the repression of gene expression and is a well-studied epigenetic mechanism in OP [56,57]. Aberrant DNA methylation patterns have been observed in genes related to bone metabolism, including those involved in the Wnt signaling pathway, osteoclast and osteogenic differentiation [58,59,60]. These epigenetic changes may contribute to the altered gene expression profiles seen in osteoporotic samples, ultimately affecting bone formation and resorption processes.

#### 2.3.2. Histone Modifications

Histone modifications, which involve the addition or removal of chemical groups to histone proteins around which DNA is wrapped, also play a critical role in regulating chromatin structure and gene expression [61,62]. Specific histone modifications have been linked to the activation or repression of genes involved in bone health [63,64]. For instance, changes in histone acetylation and methylation states have been associated with the regulation of osteoblast and osteoclast differentiation, influencing bone remodeling and the risk of OP [65,66,67].

#### 2.3.3. Non-Coding RNAs

Non-coding RNAs (ncRNAs), particularly microRNAs (miRNAs) and long non-coding RNAs (lncRNAs), have emerged as key regulators of gene expression in bone cells [68,69]. MiRNAs, short RNA molecules that bind to messenger RNAs (mRNAs) to prevent their translation into proteins, are involved in the fine-tuning of gene expression related to bone formation and resorption [70,71]. Dysregulation of specific miRNAs has been implicated in the pathogenesis of OP, providing potential targets for therapeutic intervention. Similarly, lncRNAs, which are longer RNA molecules that can regulate gene expression at multiple levels, have been shown to influence bone metabolism and are being explored as potential biomarkers and therapeutic targets in OP [72].

## 3. Transcriptomics: Understanding Gene Expression in Osteoporosis

### 3.1. Gene Expression Profiles

Transcriptomics, the study of the complete set of RNA transcripts produced by the genome under specific circumstances, provides a dynamic view of gene expression and how it is altered in diseases. Techniques such as RNA sequencing (RNA-seq) and microarrays have been instrumental in studying RNA transcripts. RNA-seq allows for high-throughput sequencing of RNA, providing comprehensive and precise information on gene expression levels, splicing events, and novel transcripts [73,74]. Microarrays, while more limited in their scope, are widely used for measuring the expression of large sets of known transcripts [75]. These techniques have greatly advanced our understanding of gene regulation, shedding light on critical genes involved in bone remodeling and metabolism [76,77,78]. In OP, several key genes involved in bone remodeling are frequently dysregulated. For instance, Runt-related transcription factor 2 (RUNX2) is a critical transcription factor necessary for both osteoblast differentiation and chondrocyte maturation. It regulates the expression of genes such as SP7, SPP1, IBSP, BGLAP2, COL1A1, COL1A2, COL10A1, and MMP13, and also influences osteoclastogenesis through the regulation of receptor activator of NF-κB (RANK) ligand (RANKL). The co-transcription factor CBFB stabilizes RUNX proteins, with CBFB2 enhancing the transcriptional activity of RUNX2, playing a crucial role in skeletal development [79,80,81]. Conversely, genes that promote osteoclastogenesis, such as RANKL and its receptor RANK, may be upregulated in OP, leading to increased bone resorption [82,83]. The RANKL/RANK signaling promotes bone turnover by enhancing resorption and leading to bone loss, while osteoprotegerin (OPG) inhibits this process by preventing the interaction between RANKL and RANK, thereby promoting bone formation. The ratio of RANKL to OPG controls the extent of osteoclastogenesis and the level of bone resorption [84,85]. An increased RANKL/OPG ratio promotes excessive osteoclast activation, leading to heightened bone resorption and decreased bone density. This imbalance diminishes the protective effects of OPG, which normally inhibits osteoclastogenesis by binding to RANKL, tipping the balance toward bone degradation. This disruption between bone formation and resorption is a hallmark of OP and is further reflected in the altered transcriptomic profiles observed in patients with the disease. Additionally, transcriptomic studies have identified changes in the expression of genes related to the ECM in OP [86]. The ECM is vital for providing structural support and maintaining bone strength and integrity. Altered expression of ECM components, including various collagens and matrix metalloproteinases (MMPs), can compromise bone quality and contribute to increased fragility [87,88,89]. Understanding these gene expression changes is essential for identifying new targets for therapeutic intervention and improving bone health in individuals at risk for OP.

Transcriptomic profiling of OP patient samples has shed light on the molecular changes linked to the disease. By comparing the gene expression profiles of OP with those from healthy controls, researchers have identified differentially expressed genes (DEGs) that may drive OP pathogenesis. Notably, there is an upregulation of genes involved in inflammation and immune responses, such as IL-6 and tumor necrosis factor-alpha (TNF-α) [90,91,92]. This chronic low-grade inflammation, prevalent in aging populations, promotes osteoclast activity, increasing bone resorption and contributing significantly to the disease. Moreover, this inflammatory response is associated with other age-related conditions, like sarcopenia, which further elevates fracture risks in the elderly [93,94]. Beyond inflammation, oxidative stress and apoptosis-related gene expression changes have also been noted in OP [95,96]. Oxidative stress, characterized by an imbalance between reactive oxygen species (ROS) and the body’s antioxidant capacity, can cause bone cell damage, contributing to OP progression [97,98]. Key antioxidant genes, including superoxide dismutase (SOD) and glutathione peroxidase (GPX), are often dysregulated in OP, leading to increased oxidative damage and apoptosis [99,100]. Furthermore, the role of apoptosis in bone health is underscored by the propagation of pro-apoptotic signals through gap junctions and hemichannels in bone cells [101,102]. These signals involve the activation of caspase family proteases, which mediate apoptosis by cleaving cellular components critical for survival. Specifically, the intrinsic pathway of apoptosis is triggered by mitochondrial dysfunction, leading to the release of cytochrome *c* and subsequent activation of caspase-9 [103,104]. The extrinsic pathway is initiated by death receptors, such as Fas and TNF receptors, which activate caspase-8 [105,106]. The propagation of these signals between osteoblasts, osteocytes, and osteoclasts disrupts bone remodeling and can lead to bone diseases such as OP and osteonecrosis. Anti-apoptotic strategies, such as targeting Bcl-2, PARP, and the PKB/Akt pathway, and utilizing treatments like bisphosphonates, estrogen therapy, and vitamin D, are proposed to prevent excessive bone cell death and manage OP [101]. These insights underscore the complex interplay between genetic, inflammatory, and oxidative stress pathways in the pathogenesis of OP.

### 3.2. Dysregulation of microRNAs and lncRNAs

As previously reported, miRNAs typically function by binding to the mRNA of target genes, inhibiting their translation into proteins. In OP, various miRNAs play essential roles in bone remodeling by modulating osteoclast and osteoblast activities [107,108]. Recent studies have reviewed miRNAs involved in osteoblast differentiation and proliferation, such as miR-185, miR-124, miR-33-5p, miR-103a, and miR-139-5p, among others. Additionally, miR-21-5p, miR-155, miR-125a-5p, miR-182, and miR-221-5p have been linked to osteoclast differentiation [108]. A systematic review identified nine dysregulated circulating miRNAs, including miR-23a-3p, miR-29a, and miR-133a, shared between OP and sarcopenia, suggesting their involvement in the pathogenesis of osteosarcopenia [109]. Clinically, miRNAs hold the potential for diagnosing, prognosing, and treating OP [110]. The OsteomiR panel, which evaluates 19 bone-specific miRNAs, is commercially available for diagnosing OP and assessing fracture risk [111]. Researchers have also identified miR-375 as a marker for high risk of PMOP and miR-203a as a diagnostic marker for fragility fractures [112]. The analysis of dysregulated miRNAs in OP patients is increasingly promising for developing miRNA-based diagnostics and therapies. For instance, miR-21, miR-24, miR-100, miR-454-3p, miR-26b-5p, and miR-584-5p were found to be upregulated, while miR-24a, miR-103-3p, and miR-142-3p were downregulated in the serum of OP patients [113,114]. These findings highlight the growing potential of miRNAs as valuable tools in managing OP. However, there are also significant challenges in translating miRNA findings into clinical applications. Developing miRNA-based diagnostics is complicated by the variability of miRNA expression across different tissues and disease stages, making it difficult to establish standardized detection protocols [115,116]. Additionally, miRNA-based therapies face hurdles such as achieving efficient and targeted delivery to bone tissue while minimizing off-target effects and ensuring the stability of miRNA molecules in the body. Overcoming these obstacles, including the natural degradation of miRNAs and ensuring their safe and specific action, remains a major barrier to their widespread clinical use.

LncRNAs play crucial roles in regulating gene expression through mechanisms such as chromatin modification, transcriptional regulation, and post-transcriptional processing. A study identified nine candidate lncRNAs in peripheral blood mononuclear cells from early postmenopausal women, highlighting MIR22HG as a notable miRNA precursor that may influence bone metabolism via the FoxO signaling pathway [117]. Additionally, transcriptome sequencing has revealed dysregulated lncRNAs and mRNAs, identifying potential hub immune-related mRNAs in OP with vertebral fractures [118]. These findings suggest that specific lncRNAs could serve as valuable biomarkers for OP and present new opportunities for therapeutic development.

### 3.3. Single-Cell RNA Sequencing

Single-cell RNA sequencing (scRNA-seq) has revolutionized the study of gene expression by allowing researchers to analyze the transcriptomes of individual cells rather than bulk tissue samples [119,120]. Unlike traditional RNA-seq, which provides an averaged gene expression profile across a population of cells, scRNA-seq offers a much higher resolution, enabling the detection of cellular heterogeneity within a tissue [119,121]. This ability to resolve gene expression at the single-cell level is crucial for understanding the diverse cell types and states present in complex tissues, such as bone. By identifying rare cell populations and revealing cell-specific responses to disease, scRNA-seq provides insights that would be missed using bulk RNA-seq, where unique transcriptomic signatures may be masked by dominant cell types [122]. This technology provides unprecedented insights into the heterogeneity of bone cells and the complex cellular interactions that underlie bone remodeling in OP [123,124]. ScRNA-seq study has revealed distinct populations of osteoblasts, osteoclasts, and osteocytes with unique gene expression profiles, indicating that these cell types are not homogenous but consist of subpopulations with specialized functions [125,126]. For example, scRNA-seq studies have identified osteoblast subpopulations that are highly active in bone formation, while others appear to be more involved in maintaining bone homeostasis [127]. Similarly, osteoclast subpopulations with varying levels of resorptive activity have been observed, suggesting that different subsets of these cells may contribute differently to bone resorption in OP [128]. Moreover, scRNA-seq has shed light on the interactions between different cell types in the bone microenvironment. For instance, it has revealed how osteocytes, the most abundant cells in bone, communicate with osteoblasts and osteoclasts through signaling molecules such as sclerostin and RANKL [129]. Understanding these cellular interactions is crucial for identifying new therapeutic targets that can modulate bone remodeling processes and prevent bone loss in OP. Table 1 summarizes the studies using scRNA-seq in OP and related traits.

## 4. Proteomics: Exploring Protein Signatures in Osteoporosis

### 4.1. Proteomic Profiling

Proteomics, the large-scale study of proteins, provides valuable insights into the molecular mechanisms underlying OP by analyzing protein expression patterns [144,145,146]. Our previous study provides a comprehensive review of the current state of proteomics in OP research [22]; thus, it will not be extensively discussed here. Unlike traditional approaches, which often fail to capture the complexity of bone metabolism, proteomic techniques provide a deeper understanding of the molecular processes that regulate bone formation and resorption. By identifying and quantifying specific proteins involved in these processes, proteomics can lead to the discovery of novel biomarkers for early diagnosis and new therapeutic targets. Differentially expressed proteins (DEPs) between healthy and OP samples can serve as biomarkers or therapeutic targets, indicating disruptions in bone metabolism and osteoblast/osteoclast function [147,148]. A recent study conducted a meta-analysis of circulating proteome data from 6430 subjects to identify proteins associated with incident hip fractures, revealing 23 proteins linked to fracture risk, including GHR, IGFBP2, GDF15, and EGFR, as well as inflammation-related proteins like CD14 and CXCL12. Pathway analysis highlighted reduced LXR/RXR activation and increased acute phase response signaling, suggesting these proteins and pathways as potential targets for future therapeutic interventions [149]. Furthermore, changes in ECM proteins, such as collagens, fibronectin, and laminins, contribute to weakened bone structure and increased fracture risk in OP [150]. Alterations in bone matrix composition, including decreased levels of type I collagen and osteonectin, along with disrupted post-translational modifications like glycosylation, phosphorylation, and cross-linking, further compromise bone quality and integrity [151,152]. Understanding these proteomic changes can facilitate the development of new strategies to enhance bone health in OP.

The clinical relevance of proteomic profiling in OP treatment cannot be overstated. By identifying and quantifying specific proteins that regulate bone metabolism, proteomics not only enhances our understanding of disease pathophysiology but also aids in developing targeted therapeutic interventions. Moreover, proteomic insights into protein expression patterns and signaling pathways provide actionable data for monitoring treatment efficacy, particularly in patients undergoing anti-resorptive therapies or anabolic treatments [22,153]. As these proteomic technologies continue to evolve, they hold promise for improving the precision and effectiveness of osteoporosis management, ultimately leading to better patient outcomes through more personalized and responsive treatment approaches.

### 4.2. Bone Turnover Markers

Bone turnover markers (BTMs) are proteins released during bone formation and resorption, reflecting the dynamic process of bone remodeling—a continuous process essential for maintaining bone health [154,155,156]. These markers, which can indicate bone resorption or bone formation, are measured through various clinical tests and are crucial for evaluating fracture risk, monitoring the effects of OP treatment, and guiding decisions on treatment duration and ‘drug holidays’ for bisphosphonates. Thus, BTMs are valuable in the clinical assessment of OP and other asymptomatic bone metabolic disorders, as traditional radiographic measures of bone mass change slowly [157]. Established BTMs, such as C-terminal telopeptide of type I collagen (CTX) and procollagen type I N-terminal propeptide (P1NP), are widely used in clinical practice to monitor bone health and the effectiveness of OP treatments [158,159]. In OP, elevated levels of CTX indicate increased bone resorption, while changes in P1NP reflect bone formation activity [160,161]. Typically, high CTX levels suggest heightened bone degradation, which correlates with an increased risk of fractures [162]. Monitoring CTX and P1NP levels during treatment helps assess therapeutic efficacy—successful anti-resorptive treatments, like bisphosphonates, typically reduce CTX levels, signaling a reduction in bone resorption, while anabolic treatments increase P1NP, indicating enhanced bone formation [163,164]. Table 2 summarizes the BTMs commonly available in clinical laboratory tests.

Proteomic studies have uncovered novel BTMs that could enhance the diagnosis and management of OP. For instance, the Osteoporotic Fractures in Men (MrOS) study utilized serum proteomics to identify 20 proteins associated with accelerated BMD loss, including CD14, SHBG, B2MG, TIMP1, and several components of the complement system (C7, C9, and CFAD) [165]. In a study of ovariectomized mice, proteomics revealed four plasma proteins significantly elevated in this OP model, with tetranectin showing a notable 50-fold increase [166]. Additionally, a proteomic analysis comparing young and old mice identified 134 proteins with age-related differences, including CDH-13, which decreases with age and has been shown to inhibit osteoclast differentiation and prevent bone loss, highlighting its potential as a BTM and therapeutic target for OP prevention [167]. Overall, the identification and validation of emerging BTMs through proteomic research hold great promise for more precise monitoring of bone turnover in OP patients, paving the way for personalized treatment strategies. By incorporating these markers into routine clinical practice, healthcare providers can tailor interventions to individual patient needs. For instance, BTMs can guide the timing and duration of bisphosphonate therapy, allowing for adjustments based on patient-specific profiles and optimizing treatment effectiveness. Furthermore, these markers are crucial in determining when to initiate or pause drug holidays, ensuring that treatments are applied efficiently and safely. The advancements in BTM research thus enable a more responsive and adaptive approach to OP management, improving patient outcomes by ensuring that treatments are both personalized and timely.

### 4.3. Key Signaling Pathways in Bone Metabolism

Proteomics provides critical insights into the signaling pathways that regulate bone metabolism and how these pathways are altered in OP. Key pathways, such as the Wnt/β-catenin pathway [168,169,170], the RANKL/RANK/OPG axis [171,172,173], and the TGF-β pathway [174,175], play central roles in maintaining the balance between bone formation and resorption. In OP, specific proteomic alterations are observed within these pathways. For instance, in the Wnt/β-catenin pathway, fibronectin 1 (FN-1) has been shown to enhance osteoblast differentiation and mineralization by interacting with integrin beta-1 (ITGB1) and activating Wnt signaling. Reduced levels of FN-1 and downregulation of ITGB1 impair osteoblast function, contributing to decreased bone formation [176]. In the RANKL/RANK/OPG axis, proteomic studies have identified an increase in RANKL expression relative to OPG, promoting osteoclastogenesis and bone resorption [177]. Modifying MSCs to silence SFRP1 can enhance their pro-osteogenic effects, improve bone regeneration by reducing the RANKL/OPG ratio, and promote osteogenic differentiation, as shown through proteomic analysis of the MSC secretome [178]. The TGF-β pathway, which influences both bone formation and resorption, is also disrupted in OP. Dysregulation of the TGF-β pathway, marked by altered levels of TGF-β1 and SMADs proteins, leads to impaired bone remodeling [174,179,180]. Studies on the secretomes of human dental pulp-derived stem cells (hDPSCs) demonstrate that TGF-β1 enhances osteogenesis and wound healing while inhibiting adipogenesis [181]. Osteoking, a Traditional Chinese Medicine (TCM), significantly enhanced BMD and bone quality in an osteoporotic fracture rat model, with proteomic analysis revealing its mechanism involves inhibiting the osteogenesis inhibitor Mgp and upregulating osteogenic markers via the TGF-β/RANKL pathway [182]. Overall, by mapping these signaling networks and their alterations at the proteomic level, researchers can identify new targets for therapeutic intervention and develop strategies to restore the balance between bone formation and resorption in OP.

## 5. Metabolomics: Metabolic Changes in Osteoporosis

Metabolomics, the comprehensive study of metabolites within biological systems, has emerged as an essential tool for understanding the metabolic alterations that occur in OP [183,184,185]. Metabolites, the small molecules involved in metabolism, provide a snapshot of an organism’s physiological state, making metabolomics a powerful approach for uncovering the biochemical pathways affected by OP. By analyzing these metabolic changes, we can identify novel biomarkers and gain deeper insights into the underlying mechanisms of the disease [186,187,188]. Figure 2 illustrates the typical workflow of metabolomics research in OP, mainly including the steps from sample collection and metabolome detection to data analysis and biological interpretation.

### 5.1. Metabolomic Profiling Techniques

Several advanced techniques are employed in metabolomic profiling to detect and quantify metabolites in biological samples such as blood, urine, and bone tissue. Nuclear Magnetic Resonance (NMR) spectroscopy is one of the key techniques, offering high reproducibility and the ability to identify a broad range of metabolites [189,190]. NMR is particularly useful for analyzing bone tissue extracts, as it can detect subtle changes in the composition of bone metabolites associated with OP [191,192]. Mass spectrometry (MS), often coupled with chromatographic techniques like liquid chromatography (LC-MS) or gas chromatography (GC-MS), is another cornerstone of metabolomic analysis [193,194,195]. MS-based approaches are highly sensitive and can detect low-abundance metabolites, making them ideal for identifying specific metabolic markers of OP [196,197,198]. Additionally, emerging techniques such as capillary electrophoresis-mass spectrometry (CE-MS) and ultra-performance liquid chromatography (UPLC) are being increasingly used in metabolomics due to their high resolution and sensitivity [199,200,201,202]. These tools are refining our understanding of the metabolic networks involved in OP by identifying numerous altered metabolites, which provide insights into the disrupted metabolic pathways in the disease and help identify novel biomarkers.

### 5.2. Key Metabolites in Bone Health

Several metabolites have been recognized for their roles in bone formation, resorption, and overall bone metabolism. For example, proline and hydroxyproline (HYP), the latter being an amino acid formed through the post-translational hydroxylation of proline, are crucial for collagen biosynthesis, structural integrity, and strength [203,204]. In PMOP patients, urinary HYP levels are significantly higher compared with those without OP, indicating increased degradation of collagen type I from the bone matrix [205,206]. Additionally, hydroxylysine (HYL), derived from the post-translational hydroxylation of lysine, exists in two forms: galactosyl hydroxylysine (GHYL) and glucosyl-galactosyl-hydroxylysine (GGHYL) [206]. The concentration of GHYL in urine has been used to evaluate fracture risk in PMOP women, both with and without fragility fractures [207]. A recent systematic review and meta-analysis identified several metabolites associated with osteopenia (ON) and OP, including elevated levels of phosphatidylcholine, galactose, and succinic acid, along with reduced levels of glycylglycine and cystine. Increased lysophosphatidylcholine levels in OP compared to healthy controls suggest its potential role as a predictor for the disease [187]. Moreover, metabolites involved in energy metabolism, such as glucose, pyruvate, and lactate, are vital for maintaining bone health [208,209,210]. Osteoblasts and osteoclasts, which are crucial for bone remodeling, have high energy demands, and disruptions in their energy metabolism can impair bone remodeling processes. In OP, there is often a shift towards anaerobic metabolism in bone cells, as indicated by increased glycolytic activity and lactate production [211]. Additionally, disruptions in the tricarboxylic acid (TCA) cycle, a key pathway for cellular energy production, are commonly observed in aging-related diseases, including OP [212]. The urea cycle, crucial for removing excess nitrogen, is also affected in OP, with metabolites such as ornithine, citrulline, and arginine being identified as potential risk factors in metabolomic-based studies [196,198,213,214].

Lipids, particularly phospholipids and sphingolipids, play a significant role in bone health [215,216]. In OP, altered lipid metabolism has been linked to reduced osteoblast function and increased osteoclast-mediated bone resorption. Specifically, increased levels of oxidized lipids and elevated plasma cholesterol disrupt bone metabolism by promoting adipogenesis and inhibiting osteoblast differentiation, which negatively impacts BMD and overall bone mass [216]. Abnormalities in fatty acid oxidation and lipid biosynthesis are also associated with bone metabolism dysfunction, contributing to an inflammatory environment that enhances osteoclast activity and leads to bone loss [217,218,219]. Studies utilizing lipidomics and metabolomics have shown significant alterations in polar and lipid metabolites, indicating disruptions in amino acid, nucleotide, and lipid metabolism in PMOP mice [220]. Research has established causal relationships between specific lipid metabolites—such as acetylcarnitine, propionylcarnitine, and myo-inositol—and heel BMD [221]. Data from the UK Biobank indicate that certain lipid metabolites are independently associated with osteoporotic fractures; for example, higher levels of degree of unsaturation and docosahexaenoic acids increase fracture risk in men, while higher levels of HDL metabolites, 3-hydroxybutyrate, and sphingomyelins decrease fracture risk in women [222]. These findings suggest that specific lipid metabolites may play distinct roles in the development and progression of OP and osteoporotic fractures.

### 5.3. Metabolomics Analyses in Osteoporosis Patients

Metabolomics has significant clinical applications in OP by enabling early diagnosis, personalized treatment, and ongoing monitoring of therapeutic efficacy [183,185,223]. Through the identification of specific metabolic signatures, metabolomics can serve as a non-invasive diagnostic tool, pinpointing individuals at high risk for fractures. Personalized treatment strategies can be developed based on an individual’s metabolic profile, targeting disrupted pathways such as lipid metabolism or energy metabolism to optimize therapeutic outcomes. Additionally, metabolomics allows for the monitoring of treatment responses by tracking metabolic changes over time, facilitating dynamic adjustments to treatment plans to enhance patient outcomes. Table 3 highlights how metabolomics can be applied to various aspects of OP management. Overall, metabolomics provides a comprehensive approach to understanding and managing the complex metabolic alterations in OP, offering new opportunities for diagnosis, treatment, and prevention.

## 6. Multi-Omics Integration: A Comprehensive Approach to Osteoporosis

### 6.1. Integrative Omics Strategies

Integrative omics, or multi-omics integration, which combines data from genomics, transcriptomics, proteomics, and metabolomics, provides a comprehensive understanding of the molecular mechanisms underlying human diseases [25,27,237,238]. By leveraging the strengths of each omics discipline, researchers can identify critical biomarkers, unravel complex biological networks, and discover novel therapeutic targets with enhanced precision. A promising strategy involves a layered analysis, where omics data from different levels are integrated sequentially. For example, genomic data can reveal genetic variants associated with OP risk, which can be linked to transcriptomic data to observe gene expression changes. Proteomic and metabolomic data further enrich this understanding by uncovering alterations in protein levels and metabolic pathways, shedding light on the molecular cascade from genetic predisposition to metabolic dysfunction leading to OP.

Studies have shown that BMSCs from ovariectomized rats exhibited increased osteogenic and lipogenic differentiation. This was accompanied by the identification of 205 DEPs and 2294 differentially expressed genes, predominantly involved in ECM-receptor interaction pathways. These findings, derived from multi-omics approaches (proteomics and transcriptomics), suggest that changes in ECM components may contribute to increased bone turnover, providing new insights into the pathogenesis of OP [239]. Alterations in ECM components, such as collagens and integrins, alter the structural integrity of the bone matrix and influence the activation of signaling pathways like Wnt/β-catenin and TGF-β [240,241], which are essential for bone formation and resorption. Dysregulation of these interactions can lead to an imbalance between bone formation and resorption, promoting excessive bone turnover, which is characteristic of OP. Similarly, hyperthyroidism-induced OP in rats is linked to excessive glucose metabolism, driven by the acetylation of key metabolic enzymes, as revealed through multi-omics analysis [242]. Review articles have emphasized the potential of multi-omics approaches to deepen our understanding of bone regeneration and the complex pathogenesis of OP, pointing toward the development of more targeted diagnostic and therapeutic strategies [243,244,245].

### 6.2. Systems Biology and Network Analysis

Systems biology is an interdisciplinary field that integrates quantitative molecular data with mathematical models to understand complex living systems [246,247]. By integrating data from multiple omics layers, systems biology provides a holistic view of how dynamic interactions within bone cells contribute to disease development, enabling the identification of potential intervention points. This comprehensive approach allows for the simulation of these interactions and the design of effective treatment strategies. Network analysis, a key component of systems biology, facilitates the visualization and interpretation of vast omics data by mapping relationships between genetic variants, gene expression changes, protein modifications, and metabolic shifts [248,249,250]. In OP and related research, network analysis identifies modules of co-expressed genes or proteins that are associated with specific aspects of bone biology, such as BMD and bone remodeling, thereby pinpointing critical pathways that may be disrupted in the disease [251,252]. Moreover, systems biology and network analysis can enhance the prediction of disease outcomes by integrating omics data with clinical information [253,254]. Machine learning algorithms trained on omics data can predict fracture risk or treatment response in OP patients, providing a more informed basis for clinical decision-making [255,256]. By leveraging these advanced analytical approaches, researchers and clinicians can gain deeper insights into the pathogenesis of OP and improve the precision of patient care.

### 6.3. Machine Learning in Multi-Omics Data Analysis for Osteoporosis

Machine learning (ML) techniques offer powerful tools for analyzing the vast and complex datasets generated by multi-omics technologies, which include genomics, transcriptomics, proteomics, and metabolomics [257,258]. These datasets provide molecular insights into OP, but their complexity poses challenges for traditional analysis methods. Supervised ML algorithms, such as decision trees, support vector machines (SVMs), and neural networks, are useful for predicting disease outcomes like fracture risk by identifying molecular signatures in labeled datasets [259]. These models can help uncover novel biomarker combinations, improving diagnostic accuracy and enabling early intervention in OP [260,261,262]. Unsupervised ML methods, such as clustering algorithms and principal component analysis (PCA), are also valuable for identifying hidden patterns in omics data, allowing for patient stratification and revealing distinct molecular profiles of OP patients [263,264]. Additionally, deep learning models, capable of handling high-dimensional data, are increasingly being applied in multi-omics research to integrate diverse datasets, offering comprehensive insights into OP mechanisms and identifying novel therapeutic targets [265,266]. By transforming complex data into actionable clinical insights, ML technologies play a crucial role in advancing personalized diagnostics and therapeutic interventions in OP research.

### 6.4. Challenges and Solutions

Despite significant advances in multi-omics integration for OP research, challenges such as managing the vast complexity of omics data, integrating diverse datasets with different biases, and ensuring high-quality data normalization remain. Additionally, generating robust multi-omics datasets requires large, well-characterized cohorts that account for the various factors influencing OP, such as age, gender, and comorbidities, as well as the dynamic nature of bone metabolism. To overcome these challenges, researchers are developing new computational methods and machine learning algorithms for data integration while also working towards standardizing data collection and processing protocols. Collaborative initiatives and data-sharing platforms further support these efforts by pooling resources and expertise. Ultimately, multi-omics integration provides a powerful approach to understanding the complex molecular networks involved in OP, offering the potential for identifying novel therapeutic targets and biomarkers and paving the way for personalized treatments.

## 7. Conclusions

The development of genomics, transcriptomics, proteomics, and metabolomics has significantly deepened our understanding of OP by revealing the complex interactions between genetic, epigenetic, and environmental factors that contribute to the disease. These omics technologies have identified critical biomarkers and molecular pathways involved in bone remodeling, providing valuable insights into OP’s pathophysiology and highlighting potential therapeutic targets. By combining data from various omics layers, multi-omics approaches offer a comprehensive view of the biological networks in OP, uncovering interactions and regulatory mechanisms that might be overlooked by single-omics studies. These holistic approaches are crucial for understanding how genetic variations, gene expression, and the activities of proteins and metabolites collectively influence bone health.

Despite these advances, translating these findings into clinical practice poses challenges due to the complexity of omics data and the need for standardized methods in data collection and analysis. Addressing these challenges requires sophisticated computational tools, robust bioinformatics platforms, and strong collaboration among researchers, clinicians, and data scientists. Nonetheless, the future of OP research is promising, with the potential to discover novel biomarkers for early diagnosis and to develop personalized treatment strategies. As our understanding of the molecular basis of OP continues to grow, new targeted therapies that both prevent bone loss and promote bone formation are likely to emerge, leading to better management of OP and improved outcomes for patients.

## Figures and Tables

**Figure 1 biomedicines-12-02389-f001:**
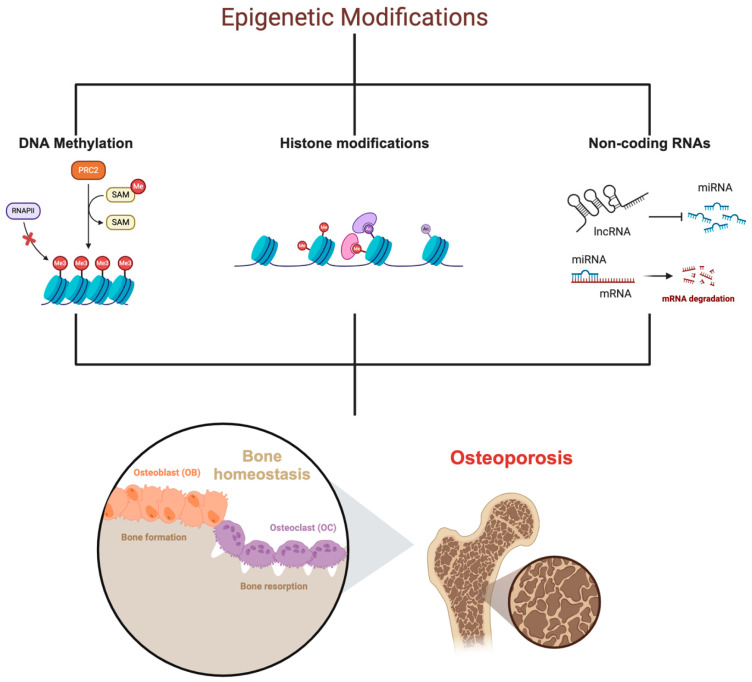
Epigenetic modifications in osteoporosis. Epigenetic modifications, including DNA methylation, histone modifications, and non-coding RNAs, regulate gene expression in bone cells. These modifications influence bone homeostasis by affecting bone formation and resorption processes, thereby contributing to the development of OP.

**Figure 2 biomedicines-12-02389-f002:**
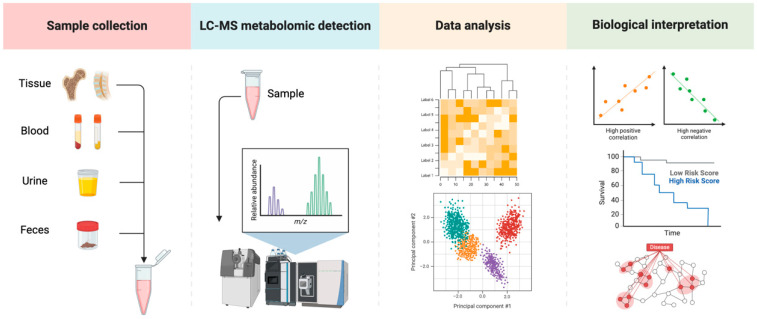
Workflow of metabolomics research in osteoporosis. The figure outlines the comprehensive workflow of metabolomics research in OP, starting with the collection of various biological samples such as bone tissue, blood, urine, and feces. Samples are then analyzed using advanced techniques to detect and quantify metabolites. The resulting data undergo thorough analysis to identify patterns, correlations, and networks, providing insights into the metabolic changes associated with OP and potentially identifying biomarkers and targets for therapeutic intervention.

**Table 1 biomedicines-12-02389-t001:** Summary of the studies using scRNA-seq in OP and related research.

Cell Types	Main Findings	Studies
Bone marrow stromal cells (BMSCs) from the femoral heads of OP patients	Reduced expression of CRIP1 in the BMSCs of OP patients is associated with decreased osteogenic capacity, and CRIP1 enhances osteogenic differentiation through the Wnt signaling pathway.	[130]
Cells from human lumbar lamina tissue of OP patients	Specific cell subgroups, such as CCL4+ NKT, CXCL8+ neutrophils, and MNDA+ macrophages, contribute to the progression of OP by creating an inflammatory environment and promoting osteoclast differentiation, while subgroups like TNFAIP6+ osteoblasts, NR4A2+ B cells, and HMGN2+ erythrocytes help maintain bone metabolism and inhibit OP through various intercellular communication pathways.	[131]
BMSCs from mice OP model	Leptin receptor-positive (LEPR+) BMSCs play a crucial role in OP pathogenesis, and the compound specnuezhenide (Spe) enhances the osteogenic potential and angiogenesis of these cells by activating the METTL3 signaling pathway.	[132]
CD14+ bone marrow monocytes of postmenopausal women	The study found that in postmenopausal OP (PMOP), there is a significant increase in CD16+ monocytes (cluster 7), which are associated with inflammation, immunity, and osteoclast differentiation, and a decrease in cluster 1 monocytes.	[133]
Granulocytes and monocytes from mouse and human blood samples	The study identified two hub genes, FAM129A and RNF24, as key players in focal adhesion related to OP. FAM129A expression was decreased in granulocytes, and RNF24 expression was increased in monocytes, with both genes influencing OP progression.	[134]
Peripheral immune cells from postmenopausal women	The study revealed an increase in myeloid cells and T cell receptor+ macrophages in PMOP, with a notable rise in natural killer cells in cases where bisphosphonate (BP) treatment failed.	[135]
A diverse range of cells from the osteoimmune microenvironment of human vertebral bone tissue	ScRNA-seq analysis delineates a complex interplay between immune cells and bone cells in the vertebral osteoimmune microenvironment, revealing distinct molecular pathways and cellular dynamics that differentiate the progression from osteoporotic vertebral compression fractures to Kümmell’s disease.	[136]
BMSCs from mice OP model	Piezo1, a mechanosensitive ion channel, plays a crucial role in bone homeostasis, and its activation by yoda1 can rescue bone loss in a hindlimb-unloading model simulating microgravity. Piezo1 activation promotes the proliferation and osteogenic differentiation of Gli1+ BMSCs via the β-catenin/ATF4 signaling pathway.	[137]
Mesenchymal stem cells (MSCs) from healthy donors	MSCs from different donors exhibit varied chromatin-accessible regulatory elements and differentiation potential into osteoblasts. The study reveals individual-specific enhancer-promoter pairs and transcriptional regulator activities, with donor 1’s MSCs showing the highest osteogenic differentiation potential.	[138]
BMSCs from diversity outbred (DO) mice	BMSCs cultured under these conditions are diverse and can be classified into mesenchymal progenitors, marrow adipogenic lineage precursors (MALPs), osteoblasts, osteocyte-like cells, and immune cells, all of which maintain transcriptomic similarity to cells isolated in vivo.	[139]
MSCs and other cells from OP patient	Twelve hub genes related to immune features are significantly associated with OP, and the expression of two specific hub genes, CDKN1A and TEFM, is greatly altered during the MSC to osteoblast transition. Additionally, chemokines and chemokine receptors are differentially enriched in various cell types, with CXCL12 being highly expressed in MSCs.	[78]
Osteoblastic lineage cells (OBCs) from the human femoral head; peripheral blood monocytes (PBMs)	Construction of immune cell and OBCs communication networks; identification of ligand-receptor (L-R) genes such as JAG1 and NOTCH1/2 that are involved in ossification; discovery of a monocyte subtype-specific subnetwork related to BMD variation in older males and postmenopausal females.	[140]
A variety of cells involved in bone metabolism	Identification of several new subtypes of macrophages and differentially over-expressed genes in the trajectory of osteoclast formation. Novel bone metabolism-related genes such as zinc finger protein 36, C3H type-like 1 (ZFP36L1), and defensin alpha 3 (DEFA3) were discovered.	[126]
Bone marrow cells from TLR9−/− mice	The study explored the alterations in myelopoiesis in the bone marrow of TLR9−/− mice. This increase in myelopoiesis contributes to inflammation-induced osteoclastogenesis and subsequent bone loss.	[141]
Osteomorphs, the daughter cells resulting from the fission of RANKL-stimulated osteoclasts	Osteomorphs are transcriptionally distinct from both osteoclasts and macrophages. They express a unique set of non-canonical osteoclast genes related to bone structure and function. Variations in the human orthologs of these osteomorph genes are linked to monogenic skeletal disorders and BMD.	[142]
Gingiva-derived mesenchymal stem cells (GMSCs)	The study identifies a distinct population of CD39+ GMSCs that play a crucial role in promoting bone formation. The study further demonstrates that CD39, produced by GMSCs, enhances osteogenesis through the activation of the Wnt/β-catenin signaling pathway.	[143]

**Table 2 biomedicines-12-02389-t002:** Bone turnover markers commonly available in clinical laboratory tests.

	BTMs	Specimen
Reflecting bone formation	Alkaline phosphatase (ALP)	Serum
	Bone-specific alkaline phosphatase (BALP)	Serum
	Procollagen type I N-terminal propeptide (P1NP)	Serum
	Procollagen type I C-terminal propeptide (P1CP)	Serum
	Osteocalcin (OC)	Serum
Reflecting bone resorption	C-terminal telopeptide of type I collagen (CTX)	Serum, urine
	N-terminal telopeptide of type I collagen (NTX)	Serum, urine
	Pyridinoline (PYD)	Urine
	Deoxypyridinoline (DPD)	Urine
	Tartrate-resistant acid phosphatase (TRACP)	Serum

**Table 3 biomedicines-12-02389-t003:** Summary of the metabolomics studies in OP patients and related research.

Samples	Techniques	Main Findings (Key Metabolites Changes)	Studies
Plasma and fecal samples from postmenopausal women	Olink proteomics and gut microbial metabolomics	Differentially abundant metabolites were related to autophagy and arginine and proline metabolism, involved in immunoinflammatory processes. Seven inflammation-related proteins showed significant differences, with CDCP1, IL10, and IL-1alph being highly discriminant for identifying PMOP.	[224]
Serum samples from OP patients and controls	Targeted amino acid metabolomics	Significant differences in amino acid metabolism were found between OP patients and healthy controls, with notable changes in the levels of 14 amino acids. Kynurenine, arginine, citrulline, and methionine were identified as potential biomarkers for OP diagnosis. Quanduzhong capsule (QDZ) treatment reversed changes in levels of 10 amino acids and influenced multiple metabolic pathways in OP patients.	[225]
Fecal samples from postmenopausal women	Untargeted metabolomics	Differences were found in the levels of indole-3-acetic acid and 7-ketodeoxycholic acid in the ON and OP groups compared to controls. Metabolite marker signatures, including these metabolites, distinguished the patients with OP from controls. Enrichment analysis revealed associations between specific metabolic pathways, such as valine, leucine, and isoleucine biosynthesis, and the microbiota biomarkers in OP.	[226]
Fecal samples from postmenopausal women	LC-MS	Key metabolites that showed significant changes between PMOP and controls included levulinic acid, N-Acetylneuraminic acid, L-pipecolic acid, and changes were noted in the alpha-Linolenic acid metabolism and selenocompound metabolism pathways.	[227]
Fecal samples from OP patients	Untargeted metabolomics	Traditional Chinese Medicine may aid OP rehabilitation by modulating gut microbiota and associated metabolic pathways. Key metabolites identified included capsazepine, Phe-Tyr, dichlorprop, D-pyroglutamic acid, and tamsulosin, which were linked to metabolic pathways such as the citrate cycle (TCA cycle) and beta-alanine metabolism.	[228]
Plasma samples from postmenopausal women	Untargeted metabolomics	The study identified 68 differentially abundant metabolites (DAMs), with most having decreased relative abundance in PMOP samples. Nine DAMs were highlighted as distinguishing factors for PMOP, including Triethanolamine, Linoleic acid, and PC(18:1(9Z)/18:1(9Z)), which showed excellent discrimination performance. Key metabolites positively correlated with bone mineral density (BMD) and T-scores included Triethanolamine, PC(18:1(9Z)/18:1(9Z)), 16-Hydroxypalmitic acid, and Palmitic acid. The candidate metabolites were primarily involved in lipid metabolism.	[229]
Serum samples from ON and OP patients and controls	Untargeted lipidomics	A total of 322 dysregulated lipid molecules were identified, with 163 upregulated and 159 downregulated in the low BMD group compared with the control. The most significantly dysregulated lipid subclasses were phosphatidylcholine (PC), triacylglycerol (TG), and phosphatidylethanolamine (PE). Other dysregulated glycerophospholipids included lysophosphatidylcholine (LPC), lysophosphatidylethanolamine (LPE), and phosphatidylinositol (PI).	[230]
Serum samples from postmenopausal women	NMR spectroscopy	Eleven metabolites, including amino acids, energy metabolism metabolites, and phospholipid precursors, showed differences between osteoarthritis (OA) patients and controls.	[231]
Serum samples from ON and OP patients and controls	Untargeted metabolomics	A total of 116 metabolites were significantly associated with low BMD. Key dysregulated metabolic pathways included histidine metabolism, aminoacyl-tRNA biosynthesis, and biosynthesis of unsaturated fatty acids.	[232]
Serum samples from OP and sarcopenia patients	CE-TOF/MS	Subjects with newly developed OP had significantly higher serum glycine (Gly) levels in the second survey. Subjects with newly developed sarcopenia had significantly lower serum taurine levels in the second survey.	[233]
Serum samples from postmenopausal women	NMR spectroscopy	Lower levels of acetate, diacylglycerol, leucine, valine, and several very low-density lipoprotein (VLDL) metabolites were observed in PMOP and atherosclerosis (AS) patients compared with controls.	[234]
Serum samples from diabetic patients with disordered bone metabolism and controls	^1^H-NMR metabolomics	Diabetic patients with disordered bone metabolism exhibited decreased levels of O-acetyl glycoprotein, proline, 1-methyl histidine, and TCA cycle products such as citric acid and α-ketoglutaric acid. In contrast, there were increased levels of branched-chain amino acids (leucine, isoleucine, valine), glucose, choline, creatine, inositol, glutamine, aspartic acid, alanine, glycine, and citrulline.	[235]
Serum samples from postmenopausal women	^1^H-NMR metabolomics	The key metabolites associated with skeletal system homeostasis and the discrimination of ON from OP included VLDL, LDL, leucine, isoleucine, allantoin, taurine, and unsaturated lipids.	[236]
Serum samples from female subjects	CE-TOF/MS	The study identified 24 metabolites with significantly different levels between pre-menopausal women with normal BMD (NN) and post-menopausal women with normal BMD (LN) and between NN and post-menopausal women with low BMD (LL). These metabolomic changes may serve as useful markers to predict estrogen deficiency and bone loss.	[214]

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
