# Peer review of "From Genomics to Metabolomics: Molecular Insights into Osteoporosis for Enhanced Diagnostic and Therapeutic Approaches"

_biomedicines, 2024, doi:10.3390/biomedicines12102389_

Round 1
Reviewer 1 Report
Comments and Suggestions for Authors
Line 22-23: The phrase “diminished bone mass” should be rephrased. You should explain what bone mass refers to (e.g., “decreased bone density and strength”).
Line 29-32: The statement regarding the highest prevalence in Africa needs more data. Are there specific reasons for this elevated prevalence in Africa, such as dietary or environmental factors?
Line 50-54: Include a phrase and say why omics technologies have been particularly impactful in OP research. How they allow for a deeper understanding of disease mechanisms at a molecular level?
Line 81-83: Explain why BMD is critical for OP risk. Add a sentence to show the role of BMD in bone strength and fracture risk.
Line 103-106: Describe more about the significance of identifying SNPs related to OP. Mention how these SNPs provide information about the molecular pathways in OP.
Line 137-140: The connection between gene-environment interactions and OP should be emphasized more. Explain how some lifestyle factors (such as diet or exercise) modulate genetic risk?
Line 181–182: Bring more information about the techniques used for studying RNA transcripts, such as RNA sequencing (RNA-seq) and microarrays.
Line 195–197: The ratio of RANKL to OPG is important in OP. Explain in more detail how the RANKL/OPG ratio changes in OP and what this suggests.
Line 220–223: You mention apoptosis-related pathways, but did not mention the specific mechanisms by which pro-apoptotic signals are propagated in bone cells. Please explain.
Line 237–238: Discuss the challenges in translating miRNA findings into clinical applications. For example, what are the difficulties in developing miRNA-based diagnostics and therapies for OP?
Line 257–258: Explain more how scRNA-seq compares to traditional RNA-seq regarding resolution and ability to detect heterogeneity within cell populations.
Lines 307–311: When discussing BTMs like CTX and P1NP, explain how their levels vary in OP and what that means for diagnosis and treatment.
Lines 330–338: The general explanation of the Wnt/β-catenin, RANKL/RANK/OPG, and TGF-β pathways needs more information about specific proteomic alterations or proteins involved.
Line 459: Add more information on how ECM-receptor interaction pathways contribute to increased bone turnover.
Line 495: Add more details about how new computational methods and machine learning algorithms address these challenges. For example, what new techniques are developed for variability in the datasets?
Comments on the Quality of English LanguageMinor language editing is needed.
Author Response
Response to Reviewer1
We thank the reviewer very much for the thorough review and valuable feedback on our manuscript. We sincerely appreciate your insightful comments, which have helped us improve the clarity and depth of our manuscript. We have integrated the comments and suggestions of all reviewers and made corresponding revisions, and the modified part is shown in red.
Line 22-23: The phrase “diminished bone mass” should be rephrased. You should explain what bone mass refers to (e.g., “decreased bone density and strength”).
Answer: We thank the reviewer for the comments, we revised the content as follows: Osteoporosis (OP) is a pervasive skeletal disorder characterized by decreased bone density and strength, along with structural deterioration, leading to increased bone fragility and a heightened risk of fractures.
Line 29-32: The statement regarding the highest prevalence in Africa needs more data. Are there specific reasons for this elevated prevalence in Africa, such as dietary or environmental factors?
Answer: We thank the reviewer for the careful review. We added the following information:
Despite abundant sunlight, the Middle East and Africa have some of the highest rickets rates globally. This could be due to the widespread occurrence of vitamin D deficiency in these regions, which may also contribute to OP[4–6]. Besides, the elevated prevalence of OP in Africa may be attributed to the continent's aging population, increased survival rates among individuals with human immunodeficiency virus (HIV), and limited access to screening and diagnostic tools, such as bone densitometry. Additionally, lack of awareness about OP, socio-economic barriers, and inadequate epidemiological data further contribute to the challenge of accurately quantifying and addressing the disease[7].
Line 50-54: Include a phrase and say why omics technologies have been particularly impactful in OP research. How they allow for a deeper understanding of disease mechanisms at a molecular level?
Answer: We thank the reviewer very much. We modified the content as follows:
In recent years, the field of OP research has been revolutionized by the advent of omics technologies, which provide powerful tools for exploring the complex molecular underpinnings of this disease. These technologies have been particularly impactful because they allow researchers to delve deeper into the molecular mechanisms that drive OP. By enabling the analysis of gene expression, protein interactions, and metabolic changes, omics approaches uncover intricate relationships between biological pathways, offering insights that were previously inaccessible.
Line 81-83: Explain why BMD is critical for OP risk. Add a sentence to show the role of BMD in bone strength and fracture risk.
Answer: We thank the reviewer for the valuable comments. We added the following information:
BMD is critical because it is a key determinant of bone strength, with lower BMD associated with an increased risk of fractures. Since BMD reflects both bone density and quality, it serves as an essential predictor of fracture risk, making its understanding vital for diagnosing and managing OP.
Line 103-106: Describe more about the significance of identifying SNPs related to OP. Mention how these SNPs provide information about the molecular pathways in OP.
Answer: We added the following content:
By pinpointing genetic variations that influence bone metabolism, SNPs help researchers map out the biological processes governing osteoblast and osteoclast function, which is invaluable for understanding disease mechanisms and developing targeted therapies for OP.
Line 137-140: The connection between gene-environment interactions and OP should be emphasized more. Explain how some lifestyle factors (such as diet or exercise) modulate genetic risk?
Answer: We thank the reviewer very much for the constructive suggestion. We added the following content:
Gene-environment interactions (G × E) [49] underscore the complexity of OP as a disease, where genetic predispositions are significantly influenced by lifestyle factors such as diet, exercise, and smoking. For example, regular physical activity can mitigate genetic risk by enhancing bone formation, while inadequate intake of calcium and vitamin D may exacerbate genetic vulnerabilities, leading to lower BMD and increased fracture risk[50,51]. These interactions highlight the need for comprehensive, personalized approaches to OP prevention and management that consider both genetic background and modifiable lifestyle factors.
Line 181–182: Bring more information about the techniques used for studying RNA transcripts, such as RNA sequencing (RNA-seq) and microarrays.
Answer: We agreed with the reviewer and added the following information:
Transcriptomics, the study of the complete set of RNA transcripts produced by the genome under specific circumstances, provides a dynamic view of gene expression and how it is altered in diseases. Techniques such as RNA sequencing (RNA-seq) and microarrays have been instrumental in studying RNA transcripts. RNA-seq allows for high-throughput sequencing of RNA, providing comprehensive and precise information on gene expression levels, splicing events, and novel transcripts[73,74]. Microarrays, while more limited in their scope, are widely used for measuring the expression of large sets of known transcripts[75]. These techniques have greatly advanced our understanding of gene regulation, shedding light on critical genes involved in bone remodeling and metabolism[76–78].
Line 195–197: The ratio of RANKL to OPG is important in OP. Explain in more detail how the RANKL/OPG ratio changes in OP and what this suggests.
Answer: We thank the reviewer for careful review and added the following content:
The ratio of RANKL to OPG controls the extent of osteoclastogenesis and the level of bone resorption[84,85]. An increased RANKL/OPG ratio promotes excessive osteoclast activation, leading to heightened bone resorption and decreased bone density. This imbalance diminishes the protective effects of OPG, which normally inhibits osteoclastogenesis by binding to RANKL, tipping the balance toward bone degradation. This disruption between bone formation and resorption is a hallmark of OP and is further reflected in the altered transcriptomic profiles observed in patients with the disease.
Line 220–223: You mention apoptosis-related pathways, but did not mention the specific mechanisms by which pro-apoptotic signals are propagated in bone cells. Please explain.
Answer: We thank the reviewer for the valuable comments, and added the following content:
Furthermore, the role of apoptosis in bone health is underscored by the propagation of pro-apoptotic signals through gap junctions and hemichannels in bone cells[101,102]. These signals involve the activation of caspase family proteases, which mediate apoptosis by cleaving cellular components critical for survival. Specifically, the intrinsic pathway of apoptosis is triggered by mitochondrial dysfunction, leading to the release of cytochrome c and subsequent activation of caspase-9[103,104]. The extrinsic pathway is initiated by death receptors, such as Fas and TNF receptors, which activate caspase-8[105,106]. The propagation of these signals between osteoblasts, osteocytes, and osteoclasts disrupts bone remodeling and can lead to bone diseases such as OP and osteonecrosis.
Line 237–238: Discuss the challenges in translating miRNA findings into clinical applications. For example, what are the difficulties in developing miRNA-based diagnostics and therapies for OP?
Answer: We thank the reviewer for the valuable suggestion, and added the following information accordingly:
However, there are also significant challenges in translating miRNA findings into clinical applications. Developing miRNA-based diagnostics is complicated by the variability of miRNA expression across different tissues and disease stages, making it difficult to establish standardized detection protocols[115,116]. Additionally, miRNA-based therapies face hurdles such as achieving efficient and targeted delivery to bone tissue, while minimizing off-target effects and ensuring the stability of miRNA molecules in the body. Overcoming these obstacles, including the natural degradation of miRNAs and ensuring their safe and specific action, remains a major barrier to their widespread clinical use.
Line 257–258: Explain more how scRNA-seq compares to traditional RNA-seq regarding resolution and ability to detect heterogeneity within cell populations.
Answer: We agree with the reviewer, and modified the content:
Single-cell RNA sequencing (scRNA-seq) has revolutionized the study of gene expression by allowing researchers to analyze the transcriptomes of individual cells, rather than bulk tissue samples[119,120]. Unlike traditional RNA-seq, which provides an averaged gene expression profile across a population of cells, scRNA-seq offers a much higher resolution, enabling the detection of cellular heterogeneity within a tissue[119,121]. This ability to resolve gene expression at the single-cell level is crucial for understanding the diverse cell types and states present in complex tissues, such as bone. By identifying rare cell populations and revealing cell-specific responses to disease, scRNA-seq provides insights that would be missed using bulk RNA-seq, where unique transcriptomic signatures may be masked by dominant cell types[122].
Lines 307–311: When discussing BTMs like CTX and P1NP, explain how their levels vary in OP and what that means for diagnosis and treatment.
Answer: We thank the reviewer and revised the content as follows:
In OP, elevated levels of CTX indicate increased bone resorption, while changes in P1NP reflect bone formation activity[160,161]. Typically, high CTX levels suggest heightened bone degradation, which correlates with an increased risk of fractures[162]. Monitoring CTX and P1NP levels during treatment helps assess therapeutic efficacy—successful anti-resorptive treatments, like bisphosphonates, typically reduce CTX levels, signaling a reduction in bone resorption, while anabolic treatments increase P1NP, indicating enhanced bone formation[163,164].
Lines 330–338: The general explanation of the Wnt/β-catenin, RANKL/RANK/OPG, and TGF-β pathways needs more information about specific proteomic alterations or proteins involved.
Line 459: Add more information on how ECM-receptor interaction pathways contribute to increased bone turnover.
Answer: We thank the reviewer for the wonderful suggestion, and revised the content as follows:
Key pathways, such as the Wnt/β-catenin pathway[168–170], the RANKL/RANK/OPG axis[171–173], and the TGF-β pathway[174,175], play central roles in maintaining the balance between bone formation and resorption. In OP, specific proteomic alterations are observed within these pathways. For instance, in the Wnt/β-catenin pathway, fibronectin 1 (FN-1) has been shown to enhance osteoblast differentiation and mineralization by interacting with integrin beta-1 (ITGB1) and activating Wnt signaling. Reduced levels of FN-1 and downregulation of ITGB1 impair osteoblast function, contributing to decreased bone formation[176]. In the RANKL/RANK/OPG axis, proteomic studies have identified an increase in RANKL expression relative to OPG, promoting osteoclastogenesis and bone resorption[177]. Modifying MSCs to silence SFRP1 can enhance their pro-osteogenic effects, improve bone regeneration by reducing the RANKL/OPG ratio, and promote osteogenic differentiation, as shown through proteomic analysis of the MSC secretome[178]. The TGF-β pathway, which influences both bone formation and resorption, is also disrupted in OP. Dysregulation of the TGF-β pathway, marked by altered levels of TGF-β1 and SMADs proteins, leads to impaired bone remodeling[174,179,180].
Line 495: Add more details about how new computational methods and machine learning algorithms address these challenges. For example, what new techniques are developed for variability in the datasets?
Answer: We thank the reviewer for the valuable suggestion, and added a dedicated subchapter titled "6.3.Machine Learning in Multi-Omics Data Analysis for Osteoporosis" to our manuscript.

Reviewer 2 Report
Comments and Suggestions for Authors
Minimal changes required see attached document
From Genomics to Metabolomics: Molecular Insights into Oste- 2 oporosis for Enhanced Diagnostic and Therapeutic Approaches
Good Introduction to incidence and prevalence of Osteoporosis and potential benefit from treatment and prevention of fractures both from an economic and quality of life measures. Describes the topic of manuscript on Genomics and metabolomics and it’s effect on osteoporosis.
Section 2 outlines concept of Genomics, predisposition and risk factors. Genomic Wide associations in 2.2 attempts to explain the single nucleotide polymorphisms, , BMD and fracture risk. The following pathays could be better explained to the reader in diagrams than text. Further associations with DNA methylation, Histone modifications and NON- coding RNA’s are appropriate.
Transcriptonomics is explained well in Section 3. Table 1 should be placed in an appendix.
Proteonomics profiling and Bone turnover markers needs to be explained further in manuscript as the clinical relevance is important for treatment of osteoporosis.
Section 5 details metabolomics extensively and is appropriate.
Conclusions drawn from manuscript are satisfactory.
References are appropriate
Happy to be accepted to publication

Author Response
Response to Reviewer2
Comments and Suggestions for Authors
Minimal changes required see attached document
From Genomics to Metabolomics: Molecular Insights into Osteoporosis for Enhanced Diagnostic and Therapeutic Approaches
Good Introduction to incidence and prevalence of Osteoporosis and potential benefit from treatment and prevention of fractures both from an economic and quality of life measures. Describes the topic of manuscript on Genomics and metabolomics and it’s effect on osteoporosis.
Section 2 outlines concept of Genomics, predisposition and risk factors. Genomic Wide associations in 2.2 attempts to explain the single nucleotide polymorphisms, BMD and fracture risk. The following pathways could be better explained to the reader in diagrams than text. Further associations with DNA methylation, Histone modifications and NON-coding RNA’s are appropriate.
Transcriptomics is explained well in Section 3. Table 1 should be placed in an appendix.
Proteomics profiling and Bone turnover markers needs to be explained further in manuscript as the clinical relevance is important for treatment of osteoporosis.
Section 5 details metabolomics extensively and is appropriate.
Conclusions drawn from manuscript are satisfactory.
References are appropriate
Happy to be accepted to publication.
Answer: We thank the reviewer very much for the thorough review of our manuscript and the positive feedback on its content. We have integrated the comments and suggestions of all reviewers and made corresponding revisions, and the modified part is shown in red.
Regarding your suggestion in Point 2. We appreciate the suggestion to use diagrams, we believe that the current text-based explanation provides clarity for the intended readers. We have ensured that the textual explanation remains accessible and coherent for readers from various backgrounds.
For Point 3, after careful consideration, we believe that keeping Table 1 within the main body of the manuscript adds immediate value to the readers by summarizing important findings in the field of transcriptomics. We hope you understanding the decision.
As per your recommendation in Point 4, we have revised the section on “4.1. Proteomic Profiling”and “4.2. Bone Turnover Markers” to further explain their clinical relevance in the treatment and management of osteoporosis.

Reviewer 3 Report
Comments and Suggestions for Authors
Li et al. Have reviewed molecular Insights into osteoporosis for enhanced diagnostic and therapeutic Approaches in aspect of the potential of multi-omics approaches including transcriptomics, proteomics and metabolomics. In the manuscript, authors introduced the state of art on the field, especially relevant for readers who are not specialists on the specific field. The manuscript is well-written and comprehensive. Even though the review articles related to the issue have been published several times, the manuscript addressed more in detail about the issue of multi-Omics Integration in osteoporosis and properly summarized the previous works as a Table of summary in each Omics section. It seems that the context of the manuscript comprehensively addresses to the goal of the present review which aims to introduce the potential of these multi-omics approaches to bridge the gap between basic research and clinical applications concerning osteoporosis, although the manuscript could not reach to discuss deep in detail about current tackles, limitations and scopes of these technologies especially integrating with clinical applications. Nevertheless, the previous works concerning the issue have been faithfully quoted and introduced in the manuscript.
Hence, the reviewer believes the manuscript acceptable for publication in Biomedicines.
Minor point)
Figure 1: It would be better to address clearer about impacts and/or corresponding keywords of results out of previous works in each three of Epigenetic Modifications, instead of the figure of osteoporosis, which leads to an immediate understanding for the impact of each modification on osteoporosis.
Author Response
Reviewer 3
Comments and Suggestions for Authors
Li et al. Have reviewed molecular Insights into osteoporosis for enhanced diagnostic and therapeutic Approaches in aspect of the potential of multi-omics approaches including transcriptomics, proteomics and metabolomics. In the manuscript, authors introduced the state of art on the field, especially relevant for readers who are not specialists on the specific field. The manuscript is well-written and comprehensive. Even though the review articles related to the issue have been published several times, the manuscript addressed more in detail about the issue of multi-Omics Integration in osteoporosis and properly summarized the previous works as a Table of summary in each Omics section. It seems that the context of the manuscript comprehensively addresses to the goal of the present review which aims to introduce the potential of these multi-omics approaches to bridge the gap between basic research and clinical applications concerning osteoporosis, although the manuscript could not reach to discuss deep in detail about current tackles, limitations and scopes of these technologies especially integrating with clinical applications. Nevertheless, the previous works concerning the issue have been faithfully quoted and introduced in the manuscript.
Hence, the reviewer believes the manuscript acceptable for publication in Biomedicines.
Minor point)
Figure 1: It would be better to address clearer about impacts and/or corresponding keywords of results out of previous works in each three of Epigenetic Modifications, instead of the figure of osteoporosis, which leads to an immediate understanding for the impact of each modification on osteoporosis.
Answer: We sincerely thank the reviewer and appreciate the positive feedback on our manuscript. We have integrated the comments and suggestions of all reviewers and made corresponding revisions, and the modified part is shown in red. Regarding your suggestion to enhance Figure 1 by clarifying the impacts or corresponding keywords of results from previous studies on epigenetic modifications, we agree that this could further facilitate understanding. However, after careful consideration, we believe that the current design of Figure 1 serves its purpose within the context of the manuscript. The figure is intended to provide a general overview of the types of epigenetic modifications and their relevance to osteoporosis. More detailed discussions regarding the specific impacts of each modification are included in the corresponding sections of the manuscript. We hope that you understand our rationale for maintaining the current figure.

Reviewer 4 Report
Comments and Suggestions for Authors
The manuscript by Li et al. is a review article summarizing recent advances in the development of improved diagnostic and therapeutic approaches using multi-omics-based technologies. In my opinion, this manuscript is timely, well written, well illustrated, and easy to follow. Nevertheless, I would like to draw the authors' attention to the fact that ML-based technologies are becoming very important for the analysis of the huge amount of data generated by multi-omics approaches. Therefore, I would like to suggest to the authors to add an appropriate subchapter discussing this issue in detail.
Author Response
Reviewer 4
Comments and Suggestions for Authors
The manuscript by Li et al. is a review article summarizing recent advances in the development of improved diagnostic and therapeutic approaches using multi-omics-based technologies. In my opinion, this manuscript is timely, well written, well illustrated, and easy to follow. Nevertheless, I would like to draw the authors' attention to the fact that ML-based technologies are becoming very important for the analysis of the huge amount of data generated by multi-omics approaches. Therefore, I would like to suggest to the authors to add an appropriate subchapter discussing this issue in detail.
Answer: We thank the reviewer for the insightful comments and for acknowledging the strengths of our manuscript. We have integrated the comments and suggestions of all reviewers and made corresponding revisions, and the modified part is shown in red. We fully agree that ML-based methods are increasingly critical for handling and interpreting the complexity of these datasets. In response to the suggestion, we have added a dedicated subchapter titled "6.3.Machine Learning in Multi-Omics Data Analysis for Osteoporosis" to our manuscript. This section discusses the various ML techniques, including supervised and unsupervised learning, used to identify patterns, classify data, and predict outcomes from multi-omics studies. We also highlight the importance of integrating ML into clinical decision-making, offering predictive models for osteoporosis diagnosis and treatment.
